# A network-centric approach for estimating trust between open source software developers

**Hitesh Sapkota**[1], **Pradeep K. Murukannaiah**[2], **Yi Wang**[1]

**1** Software Engineering, Rochester Institute of Technology, Rochester, NY, United States of America,
**2** Intelligent Systems-EWI, Delft University of Technology, Delft, The Netherlands

\* P.K.Murukannaiah@tudelft.nl

**Data Availability Statement:** All data supporting the findings in this paper are openly available at https://doi.org/10.5281/zenodo.3522461.

**Funding:** The author(s) received no specific funding for this work.

## Abstract

Trust between developers influences the success of open source software (OSS) projects. Although existing research recognizes the importance of trust, there is a lack of an effective and scalable computational method to measure trust in an OSS community. Consequently, OSS project members must rely on subjective inferences based on fragile and incomplete information for trust-related decision making. We propose an automated approach to assist a developer in identifying the trustworthiness of another developer. Our two-fold approach, first, computes *direct* trust between developer pairs who have interacted previously by analyzing their interactions via natural language processing. Second, we infer *indirect* trust between developers who have not interacted previously by constructing a community-wide developer network and propagating trust in the network. A large-scale evaluation of our approach on a GitHub dataset consisting of 24,315 developers shows that contributions from trusted developers are more likely to be accepted to a project compared to contributions from developers who are distrusted or lacking trust from project members. Further, we develop a pull request classifier that exploits trust metrics to effectively predict the likelihood of a pull request being accepted to a project, demonstrating the practical utility of our approach.

## 1 Introduction

Trust is a critical factor for enabling effective online collaboration in open source software (OSS) project teams [1]. OSS team members are more likely to collaborate, share knowledge, and accept others' contributions when they trust each other [2]. Trustworthiness also accelerates new member recruitment [3, 4], and, consequently, brings innovative ideas and work procedures to a project [5].

The importance of trust in OSS development has been long known. Extant research on trust in software engineering focuses on small-scale empirical inquiries aiming to explain the antecedence and consequence of trust (or lack of it) in software engineering teams [6–9], and mechanisms to help build and maintain trust [10, 11]. However, the research community lacks

**Competing interests:** The authors have declared that no competing interests exist.

an automated and scalable approach to assess trust among developers, particularly those who have not directly interacted. Thus, to make decisions on whether an individual is trustworthy, an OSS practitioner has to make several subjective inferences based on fragile and incomplete information dispersed in multiple repositories [3, 12]. Also, such information is often not readily available, and identifying it (from many noisy sources) requires substantial manual labor [13].

We propose an automated approach for estimating trust between developers in an OSS community. Our network-centric approach involves three key steps. It (1) constructs a community-wide developer network, utilizing various social coding traces from a community [3, 14, 15]; (2) analyzes interactions between pairs of developers directly connected in the network to estimate trust between them; and (3) employs well-known trust propagation methods [16] to estimate indirect trust between pairs of developers connected in the network by at least one path. Thus, our approach can be used to estimate trust between two members of a project as well as a member of a project and a newcomer (potential contributor).

We perform extensive experiments, driven by two research questions ($RQ_1$ and $RQ_2$), to empirically evaluate and demonstrate the practical utility of our approach.

Since we propose a computational method for estimating trust, $RQ_1$ seeks to evaluate the accuracy of the estimates our approach yields. This is the foundation for applying our approach in practical software engineering use cases.

$RQ_1$. How effective is the proposed network-centric approach for estimating direct and indirect trust between developers in an OSS community?

Prior literature shows that trust is a key factor in determining how a developer's contribution to a project is evaluated. For instance, Sinha et al. [17] and Gousios et al. [18] identify that trust between a new developer and the members of a project is a significant factor in determining whether the new developer's contribution to a project is accepted or not, implying that contributions from more trusted developers are more likely to be accepted. $RQ_2$ seeks to empirically evaluate this observation and demonstrate the practical utility of our trust computation model.

$RQ_2$. How effective are the trust metrics computed from our approach in determining whether a contribution is accepted to or rejected from a project?

We investigate these research questions via an innovative empirical study with historical data from 179 Python projects on GitHub. All these projects adopt the pull request model [19], representing a community of Python developers. We construct a network for this community consisting of 24,315 unique developers.

We find that our methods to estimate both direct trust (based on developer interaction analysis) and indirect trust (based on trust propagation) are effective on the GitHub Python developers network. Further, we find that the higher the computed trust between a new developer and the members of a project, the higher the likelihood of the developer's contribution (pull request) being accepted to the project. Thus, the proposed approach is valid, and the trust values it computes are useful for supporting various decision scenarios, including setting proper expectations [2] and evaluating pull requests [14] in the OSS development process.

## Contributions

- A novel network-centric approach to help OSS practitioners in automatically evaluating the trustworthiness of other developers regardless of whether the developer have directly interacted or not in the past.

- An empirical evaluation demonstrating the validity and utility of our approach.

- Open source software and dataset of 179 Python projects (including annotated pull requests) [20], which can be used to construct a developer network, and estimate and validate trust between developers.

## Organization

Section 2 describes the preliminaries required to understand our approach. Section 3 presents our approach. Section 4 describes the evaluation design. Section 5 reports and discusses results. Section 6 reviews the related work. Section 7 concludes the paper.

## 2 Preliminaries

We define trust and describe a computational model of trust.

### 2.1 Defining trust

Trust has been studied in many disciplines and may have a different meaning in each context. We adopt Golbeck's [21] definition: *Alice trusts Bob if she commits to an action based on the belief that Bob's future actions will lead to a good outcome.* This definition is widely used in online social collaboration (our setting) and is easy to incorporate within a computational framework (our objective).

We adapt Golbeck's definition to our setting as follows:

1. Alice accepts Bob's contribution to a project if she trusts him.

2. Alice trusts Bob if she believes that Bob's future actions (e.g., maintaining the code he contributed, assisting developers depending on his code, and so on) will lead to the success of the project.

### 2.2 Modeling trust

We model trust based on Jøsang's subjective logic [22], which in turn is derived from Dempster-Shafer theory [23]. Jøsang represents trust in terms of *belief* ($\mathcal{B}$), *disbelief* ($\mathcal{D}$), and *uncertainty* ($\mathcal{U}$). To understand the intuitions behind trust parameters, consider an example proposition from Alice to Bob that "*Charlie is a great Python developer.*" This proposition reflects Alice's opinion of Charlie. Hearing Alice's opinion, Bob may "believe" that Charlie is indeed a good Python developer, but Bob may be "uncertain" about it. Next, consider that Bob hears from Dorothy that Charlie fixed a nontrivial bug in her Python project. This reduces Bob's uncertainty in his belief about Charlie being a great Python developer. In essence, as Table 1 shows, a trustor's belief in an opinion (about a trustee) represents the trustor's tendency to believe the opinion, disbelief represents the tendency to disbelieve the opinion, and uncertainty represents the trustor's confidence (or lack of it) in the belief and the disbelief.

Table 1. Trust as a function of belief, disbelief, and uncertainty.

| Belief | Disbelief | Uncertainty | Interpretation |
|---|---|---|---|
| High | Low | Low | Trust |
| Low | High | Low | Distrust |
| — | — | High | Lack of trust |

In the classic formulation of subjective logic, each trust parameter can take a value in the range [0, 1]. However, the three trust parameters must always add up to one. Thus, the value one trust parameter can take is constrained by the values of the other two trust parameters. When uncertainty takes the zero value, the opinion is considered *dogmatic*. In contrast, when uncertainty takes the unity value (which happens when there is no evidence to infer trust), the opinion is considered *vacuous*.

### 2.3 Computing trust

Our computational model of trust operates in two scenarios.

**Direct trust**. If two developers have a history of interactions, we compute direct trust between them by analyzing their interactions. Specifically, we compute positive and negative *evidence* from their interactions and systematically map the evidence to the three trust parameters (belief, disbelief, and uncertainty).

**Indirect trust**.If two developers have not interacted so far but are connected in the developer network via a chain of other developers, we propagate trust. The subjective logic defines two operators for propagation: *transitivity* and *cumulative fusion* [16].

- Given two developers connected via a chain, the transitivity operator computes trust between the two developers by discounting belief increasing the uncertainty as the length of a chain increases.

- Given multiple chains between two developers, the fusion operator combines trust from each chain so as to amplify belief or disbelief, and reduce uncertainty.

## 3 Approach

Fig 1 shows the three key steps in our approach. First, we gather information related to developers and projects, and construct a community-wide developer network (CDN). Second, we

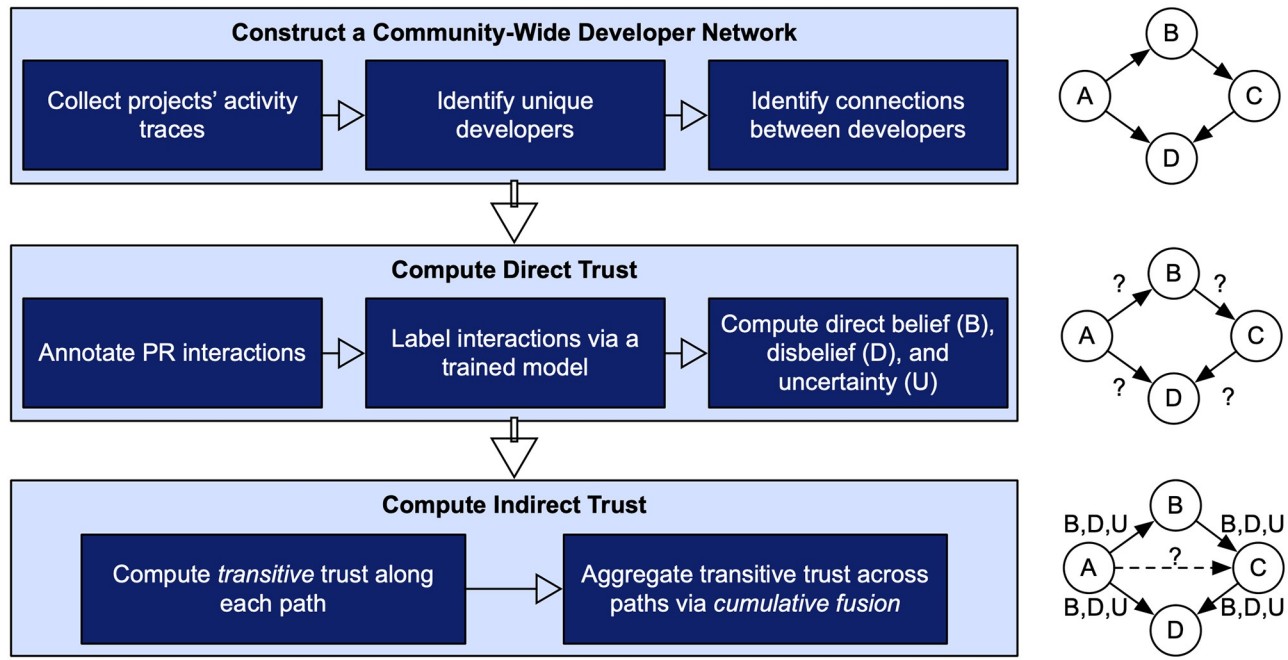

**Fig 1. Three key steps in the proposed approach for estimating trust.**

compute trust between pairs of directly connected developers in the CDN. Finally, we compute trust between pairs of developers indirectly connected in the CDN.

### 3.1 Community-wide developer network

We are interested in estimating the trustworthiness of a potential contributor to an OSS project. Accordingly, we construct a CDN, which provides valuable information about collaboration between developers in an OSS community [24]. The CDN represents a community of developers from multiple OSS projects sharing some common characteristics (which, in our case, is using same programming language).

We define a CDN as a weighted, directed graph constructed from developers' activity traces such that each (1) *node* in the graph represents a unique developer, (2) *directed edge* from one developer (source) to another (target) represents the source developer's direct trust toward the target developer, and (3) edge includes three *weights*, representing the three trust parameters— belief, disbelief, and uncertainty.

We consider interactions between developers in OSS projects for constructing the CDN. However, our approach for computing direct and indirect trust is generic in that additional factors can be easily incorporated into it.

The process of constructing the CDN consists of the following steps.

1. We identify a set of OSS repositories that share a common contribution acceptance mechanism. Specifically, we take a sample of OSS Python projects that adopt the pull request model, forming a community of Python developers.

2. For each repository in the set, we collect: (a) the developers in the repository, (b) the pull requests made to that repository (by developers), and (c) the comments associated with each pull request.

3. We add each unique developer in the community as a node to the CDN. We add a directed edge from developer *A* to developer *B* if *A* has commented on at least one pull request generated by *B*.

Our objective is to assist an OSS developer, say Alice (a project owner), in evaluating the trustworthiness of another OSS developer, say Charlie (a potential contributor to Alice's project). Our approach computes Alice's trust toward Charlie in one or two steps depending on the scenario.

- If there is a directed edge from Alice to Charlie in the CDN, we compute direct trust between them in one step (Section 3.2).

- If Alice and Charlie are not directly connected but there exists a directed path between them in the CDN, we compute trust in two steps (Sections 3.2 and 3.3).

- If there is no path between Alice and Charlie in the CDN (which can happen when the CDN is disconnected), we do not offer any insight on trust between them.

### 3.2 Direct trust computation

Recall that a directed edge from a developer *A* to a developer *B* in the CDN indicates that *A* has commented on at least one contribution submitted by *B*. We compute the direct trust of *A* toward *B* based on *A*'s opinions on the contributions submitted by *B*.

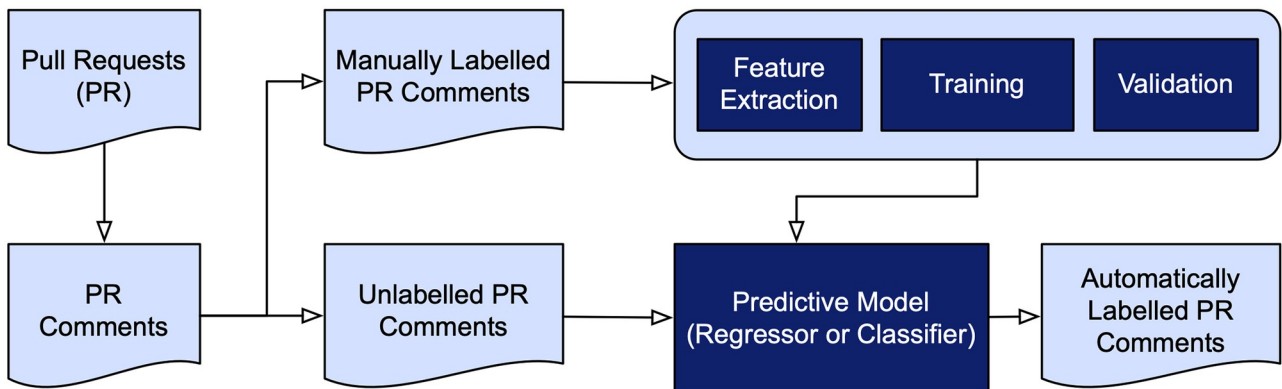

**Fig 2. The steps we follow to train and validate a predictive model for categorizing an evaluator's opinion toward a contributor's pull request as likely to increase, decrease, or not affect trust.** We experiment with both regression and classification models.

In essence, if *A* finds a contribution from *B* as valuable to the project, *A*'s trust toward *B* increases (i.e., *belief* increases). In contrast, if *A* finds *B*'s contribution as buggy or unnecessary, *A*'s trust toward *B* reduces (i.e., *disbelief* increases). Further, trust builds progressively as *A* comments on more contributions from *B* (i.e., *belief* and *disbelief* are updated after *A* comments on each of *B*'s contributions, and *uncertainty* reduces as the number of such comments increases).

We employ a predictive (regression or classification) model to infer whether *A*'s trust toward *B* is likely to increase, decrease, or not change based on *A*'s comments on *B*'s contribution. Fig 2 shows the steps we follow to train and validate the predictive model.

1. We manually annotate a set of contributions (Section 3.2.1). Specifically, we treat *A*'s comments on a specific contribution by *B* as an opinion *A* expresses toward *B* with respect to the contribution. We label each such opinion as (1) strongly positive, (2) weakly positive, (3) neutral, (4) weakly negative, or (5) strongly negative, where positive, negative, and neutral opinions indicate that *A*'s trust toward *B* is likely to increase, decrease, or be unaffected, respectively, according to the specific comments.

2. We extract four types of features from each annotated comment (Section 3.2.2).

3. We train a predictive model on the annotated comments (Section 3.2.3). The model predicts *A*'s opinion on *B* based on the features extracted from *A*'s comments on a contribution by *B*.

4. We aggregate *A*'s opinion on *B* across contributions and map the aggregate opinion to the trust parameters—belief, disbelief, and uncertainty (Section 3.2.4).

**3.2.1 Annotation.** Table 2 shows a few examples for different types of opinions a developer may have toward a contribution based on the comments made by the developer. We treat a commenter's opinion toward a contribution as (1) positive if the commenter appreciates the contribution, (2) negative if the commenter discourages the contribution, or (3) neutral if the commenter neither appreciates nor discourages the contribution. Within the positive and negative opinions, the strong and weak designations are based on the extent to which a comment is appreciative or discouraging.

We take all pull requests in our dataset and extract all comments from each pull request. We randomly pick a subset of the pull requests for annotation. Suppose that a pull request

**Table 2. Examples of developer comments on pull requests and annotator's opinion on those comments.**

| Annotator's opinion | Developer's comment |
|---|---|
| Strongly positive | • *Wow, amazing work. Thanks!* <br> • *Great work @username, I like the example. All of my comments were nits/stylistic.* |
| Weakly positive | • *Thanks! @username can you confirm whether the reports are already properly handling location restrictions?* <br> • *Looks reasonable, can you add a whatsnew note for 0.23.4 (bug fixes)?* |
| Neutral | • *@username can you provide us some directions about how to reproduce the initial issue?* <br> • *This is done in 1288e65. Thanks.* |
| Weakly negative | • *I dislike this honestly. Just do the rework and then replace the existing shadowling with it, don't remove it, then start a rework which may or may not ever get done.* <br> • *I agree with @username that a test is needed for this new feature.* |
| Strongly negative | • *Screaming out your only reason to remove a functionality does not make it more valid.* <br> • *Awful. We are never getting that replacement, you know.* |

generated by developer *B* is picked for annotation and that a developer *A* has commented on this pull request. Considering all comments *A* made about the picked pull request, we label *A*'s opinion on *B* on a five-point scale: 5 (strongly positive), 4 (weakly positive), 3 (neutral), 2 (weakly negative), 1 (strongly negative). If more than one developer commented on the picked pull request, we annotate the opinion of each of those developers toward *B*, considering the interactions between the developer and *B*.

The annotation was done in three phases involving two Software Engineering graduate student researchers as annotators. During the first phase, both annotators rated 50 pull requests (randomly picked), independently. After labeling, we computed the intraclass correlation coefficient (ICC), a commonly-used interrater reliability (IRR) metric for ordinal data [25]. The ICC for the first phase was 0.88, which is considered to be excellent [25]. To make sure that the labelling process is reliable, in the second phase, both annotators labelled another set of 50 pull requests (randomly picked), independently. The ICC for the second phase was 0.82. Since the ICC was sufficiently high in the first two phases, in the third phase, one annotator labelled 200 pull requests, and another annotator labelled a different set of 100 pull requests.

As a result of the annotation process, a total of 400 pull requests were labelled, including interactions between 616 developer pairs (a pair includes a pull request generator and an evaluator) and of 702 comments. Note that a pull request can involve more than one interaction since more than one evaluator may comment on the pull request. Further, an evaluator may comment on a pull request more than once.

**3.2.2 Feature extraction.** We extract four types of features from the labelled data to train regression models.

*Word Embedding.* We use Google Word2Vec [26] to vectorize each comment. Instead of using a pre-trained model, we train our own Word2Vec model on software engineering data because a domain-specific model may have better semantic representation compared to the pre-trained generic model. To train the model, we employ all pull request comments corresponding to the training dataset used for CDN construction (Section 4.1). We use the trained model to get a 300-dimensional vector for each word in a sentence. Finally, we take the mean of word vectors in a comment to get the vector representation of the comment.

*Sentiment.* Sentiment expressed in a developer's comment is likely to be an indicator of the commenter's opinion toward the contribution. We employ SentiStrength-SE [27], a software engineering lexicon, for extracting positive and negative sentiment scores.

*Social.* The strength of the social connection between two developers can influence the trust between them. Accordingly, we include: (1) a binary variable indicating whether the pull requester follows commenter, (2) a binary variable indicating whether commenter follows the pull requester, (3) the number of projects shared between commenter and pull requester, (4) the number of conversations between commenter and pull requester in pull requests, and (5) Two integer values, in the range 0–6, indicating the roles of pull request generator and commenter in a project. The role can be owner, member, collaborator, contributor, first-time contributor, first-timer or none [28]. We assign an integer value for each role in descending order with owner as 6 and none as 0.

*Contributions.* The contribution-related features we include are the: (1) total number of comments in the contribution, (2) maximum, minimum, standard deviation, and mean length of the comments, (3) minimum, maximum, standard deviation, and mean time between consecutive conversations, (4) number of files changed in the contribution, and (5) the number of lines added and deleted.

**3.2.3 Opinion prediction.** Considering the large number of developer pairs and interactions between them, it is not feasible to manually assign an opinion for each interaction. Thus, in our approach, first, experts manually assign labels to a small subset of developer interactions. Then, we train an automated technique on the expert-annotated interactions to predict the opinion labels for the remaining interactions.

The opinion prediction problem can be addressed via regression, where the predicted opinion is a continuous value in the range [1, 5], or via classification, where the predicted opinion is one of the discrete values in the set {1, 2, 3, 4, 5}. We experiment with opinion prediction via regression as well as classification. We employ each of the 616 interactions (Section 3.2.1, last para) annotated with an opinion label as an observation in training and testing the opinion prediction techniques.

We experiment with five techniques for opinion prediction: (1) XGBoost [29], (2) AdaBoost [30], (3) Bagging [31], (4) Lasso [32], and (5) Support Vector Machines (SVM) [33], employing the ScikitLearn implementation for each technique. Each of these techniques can be used for regression as well as classification [29–33].

We compare these prediction techniques via Mean Absolute Error (MAE) and employ the best performing technique for the automated labeling task (Section 5.1). Given an evaluator's comments on a contributor's pull request, the automated (regression or classification) technique predicts the evaluator's opinion of the contributor as a value in the range: 1 (strongly negative) to 5 (strongly positive).

**3.2.4 Opinion aggregation and trust mapping.** The opinion prediction technique above predicts a developer *A*'s opinion on a developer *B*, considering *A*'s comments on a specific contribution by *B*. However, to compute *A*'s trust toward *B*, we must aggregate all opinions of *A* toward *B*. To do so:

1. We gather all of *B*'s contributions on which *A* has commented.

2. We predict *A*'s opinion of *B* for each contribution gathered above.

3. We compute two scores *r* and *s* such that:

   a. *r* is the sum of all positive opinion values, where each weakly positive opinion is counted as 0.5 and each strongly positive opinion is counted as 1; and,

   b. *s* is the sum of all negative opinion values, where each weakly negative opinion is counted as 0.5 and each strongly negative opinion is counted as 1.

4. Finally, given $r$ and $s$, we employ the heuristics suggested by Jøsang [16] to compute trust dimensions as follows:

$$\mathcal{B} = \frac{r}{r+s+2}; \quad \mathcal{D} = \frac{s}{r+s+2}; \quad \mathcal{U} = \frac{2}{r+s+2} \tag{1}$$

**Example 1** *Suppose that Alice has commented on two of Charlie's contributions to a project and three of Charlie's contributions to another project. Alice's opinion on Charlie's contributions to the first project were weakly negative and weakly positive. Alice's opinion on Charlie's contribution to the second project were weakly negative, strongly positive, and strongly positive. Compute Alice's trust toward Charlie.*

Let $\mathcal{L}_{neg} = \{$weakly negative, weakly negative$\}$, and $\mathcal{L}_{pos} = \{$weakly positive, strongly positive, strongly positive$\}$ be the set of negative and positive opinions, respectively, aggregated across the five contributions spanning two projects. Then,

$$r \quad = 0.5 + 1 + 1 = 2.5; \quad s = 0.5 + 0.5 = 1.$$
$$\mathcal{B}_{\text{Charlie}}^{\text{Alice}} \quad = 2.5/5.5 = 0.45; \quad \mathcal{D}_{\text{Charlie}}^{\text{Alice}} = 1/5.5 = 0.18; \quad \mathcal{U}_{\text{Charlie}}^{\text{Alice}} = 2/5.5 = 0.36.$$

### 3.3 Indirect trust computation

If a developer $A$ has never interacted with a developer $C$ (i.e., in our case, $A$ never commented on any of $C$'s pull requests), we would not have any clue to estimate direct trust from $A$ to $C$. However, there might be some indirect evidence. For instance, $A$ and $C$ may have a common collaborator $B$ such that $A$ has an opinion on $B$, and $B$ has an opinion on $C$. In such a scenario, we estimate $A$'s trust toward $C$ via *trust propagation*.

Given that $A$ is connected to $C$ via at least one path in the CDN, we compute $A$'s indirect trust toward $C$ as follows.

1. We select a set of paths from $A$ to $C$ via a length cutoff (Section 3.3.1).

2. We compute direct trust between each pair of directly connected developers on each selected path (Section 3.2).

3. We propagate trust along each selected path (Section 3.3.2).

4. We aggregate trust across all selected paths (Section 3.3.3).

Since the sum of the three trust parameters (belief, disbelief, and uncertainty) is one, if we know two parameters, we can compute the third. Therefore, we only describe the computation of two trust parameters (belief and uncertainty) in the rest of this section.

**3.3.1 Path length cutoff.** The accuracy of indirect trust estimation depends on the paths we choose for propagation [34, 35]. In a CDN, without any *length* restrictions, there can be a large number of paths for propagating trust from $A$ to $C$. Golbeck [21] compares the accuracy of indirect trust computation against the length *cutoff* chosen for selecting paths (the cutoff defines the maximum length of paths included in propagating trust). Based on empirical evaluations on different trust networks, Golbeck observes that: (1) a higher cutoff includes more paths but decreases the accuracy of indirect trust estimation; and (2) a lower cutoff causes the loss of trust chains for many pairs of nodes.

Our approach, too, employs a length cutoff to select paths. We empirically tune the cutoff value for our CDN (Section 5.1). However, when there is no path of length less than or equal

to cutoff between a pair of nodes, we consider all possible shortest paths between those nodes to maximize connectivity.

**3.3.2 Trust propagation through a single path.** There are several strategies to propagate trust along a path [16, 36]. We describe and empirically evaluate (Section 5.1) two commonly used strategies.

Suppose there are three developers $A$, $B$, and $C$ and $A \rightarrow B \rightarrow C$ is a trust path in the CDN. Further, suppose that $\{\mathcal{B}_B^A, \mathcal{U}_B^A\}$ and $\{\mathcal{B}_C^B, \mathcal{U}_C^B\}$ are direct belief and uncertainty between $A \rightarrow B$ and $B \rightarrow C$, respectively. Then, we compute $A$'s indirect trust toward $C$ in one of the following ways.

- The *TP-Minimum* strategy propagates the trust along a path based on the trust corresponding to the link with the weakest trust along the path. That is:

$$
\begin{aligned}
\mathcal{B}_C^A &= \min\left(\mathcal{B}_B^A, \ \mathcal{B}_C^B\right) = \mathcal{B}_{\min} \\
\mathcal{U}_C^A &= \max\left(\mathcal{U}_i \text{ where } \mathcal{B}_i = \mathcal{B}_{\min}\right)
\end{aligned}
\tag{2}
$$

- The *TP-Discount* strategy propagates trust by successively discounting trust values (reducing belief and increasing uncertainty) along the path. That is:

$$
\begin{aligned}
\mathcal{B}_C^A &= \mathcal{B}_B^A \times \mathcal{B}_C^B \\
\mathcal{U}_C^A &= 1 - \mathcal{B}_B^A(1 - \mathcal{U}_C^B)
\end{aligned}
\tag{3}
$$

**3.3.3 Trust aggregation across multiple paths.** There can be multiple trust paths between two developers in the CDN. After propagating trust along each path, we compute indirect trust between the developers by aggregating trust values computed across all paths. Similar to trust propagation along a single path, there are multiple strategies for aggregating trust values [36]. We describe and empirically evaluate (Section 5.1) three popular strategies.

Suppose that $A$, $B$, $C$, and $D$ are four developers, and that there exist two trust paths from $A$ and $D$ along $A \rightarrow B \rightarrow D$ and $A \rightarrow C \rightarrow D$. Let $\{\mathcal{B}_D^{A:B}, \mathcal{U}_D^{A:B}\}$ and $\{\mathcal{B}_D^{A:C}, \mathcal{U}_D^{A:C}\}$ be propagated belief and uncertainty values along $A \rightarrow B \rightarrow D$ and $A \rightarrow C \rightarrow D$, respectively. We compute the aggregate belief ($\mathcal{B}_D^A$) and uncertainty ($\mathcal{U}_D^A$) from $A$ toward $D$ via one of the following strategies.

- The *AP-Mean* strategy computes aggregate trust as the mean of trust values propagated on each path. That is:

$$
\begin{aligned}
\mathcal{B}_D^A &= \frac{1}{2}\left(\mathcal{B}_D^{A:B} + \mathcal{B}_D^{A:C}\right) \\
\mathcal{U}_D^A &= \sqrt{\left(\mathcal{U}_D^{A:B}\right)^2 + \left(\mathcal{U}_D^{A:C}\right)^2}
\end{aligned}
\tag{4}
$$

- The *AP-Maximum* strategy computes the aggregate trust values by choosing the maximum transitive trust propagated on each path. That is:

$$
\begin{aligned}
\mathcal{B}_D^A &= \max\left(\mathcal{B}_D^{A:B}, \ \mathcal{B}_D^{A:C}\right) = \mathcal{B}_{\max} \\
\mathcal{U}_D^A &= \min\left(\mathcal{U}_i \text{ where } \mathcal{B}_i = \mathcal{B}_{\max}\right)
\end{aligned}
\tag{5}
$$

- The *AP-Consensus* strategy computes the aggregate trust values by the fusion of transitive trust propagated on each path. That is:

  - Case 1: $\mathcal{U}_D^{A:B} + \mathcal{U}_D^{A:C} - \mathcal{U}_D^{A:B} \times \mathcal{U}_D^{A:C} \neq 0$

$$
\begin{aligned}
\mathcal{B}_D^A &= (\mathcal{B}_D^{A:B} \times \mathcal{U}_D^{A:C} + \mathcal{B}_D^{A:C} \times \mathcal{U}_D^{A:B})/(\mathcal{U}_D^{A:C} + \mathcal{U}_D^{A:B} - \mathcal{U}_D^{A:B} \times \mathcal{U}_D^{A:C}) \\
\mathcal{U}_D^A &= (\mathcal{U}_D^{A:C} \times \mathcal{U}_D^{A:B})/(\mathcal{U}_D^{A:C} + \mathcal{U}_D^{A:B} - \mathcal{U}_D^{A:B} \times \mathcal{U}_D^{A:C})
\end{aligned}
\tag{6}
$$

  - Case 2: $\mathcal{U}_D^{A:B} + \mathcal{U}_D^{A:C} - \mathcal{U}_D^{A:B} \times \mathcal{U}_D^{A:C} = 0$

$$
\begin{aligned}
\mathcal{B}_D^A &= (\gamma^{B/C} \times \mathcal{B}_D^B + \mathcal{B}_D^C)/(\gamma^{B/C} + 1) \\
\mathcal{U}_D^A &= 0
\end{aligned}
\tag{7}
$$

$$
\text{where,} \ \gamma^{B/C} = \lim (\mathcal{U}_D^C/\mathcal{U}_D^B)
$$

**Example 2** *Suppose that A and F are two developers in a CDN, who have not interacted directly. F submits a contribution to A's project, and the A wants to estimate F's trustworthiness. Although A and F are not directly connected, there exist two paths of length $\leq 3$ (cutoff) between A and F. Let the direct trust values along each path be as shown in* Fig 3. *Compute A's indirect trust toward F, considering different trust propagation and aggregation strategies.*

- Considering *TP-Minimum* for propagation:

$$
\begin{aligned}
\mathcal{B}_F^{A:B:C} &= \min (\mathcal{B}_B^A, \ \mathcal{B}_C^B, \ \mathcal{B}_F^C) = 0.3 \\
\mathcal{U}_F^{A:B:C} &= \max (\mathcal{U}_C^B, \ \mathcal{U}_F^C) = 0.5 \\
\mathcal{T}_F^{A:B:C} &= (\mathcal{B}_F^{A:B:C}, \ \mathcal{U}_F^{A:B:C}) = (0.3, 0.5) \\
\mathcal{B}_F^{A:D:E} &= \min (\mathcal{B}_D^A, \ \mathcal{B}_E^D, \ \mathcal{B}_F^E) = 0.3 \\
\mathcal{U}_F^{A:D:E} &= \mathcal{U}_D^A = 0.2 \\
\mathcal{T}_F^{A:D:E} &= (\mathcal{B}_F^{A:D:E}, \ \mathcal{U}_F^{A:D:E}) = (0.3, 0.2)
\end{aligned}
$$

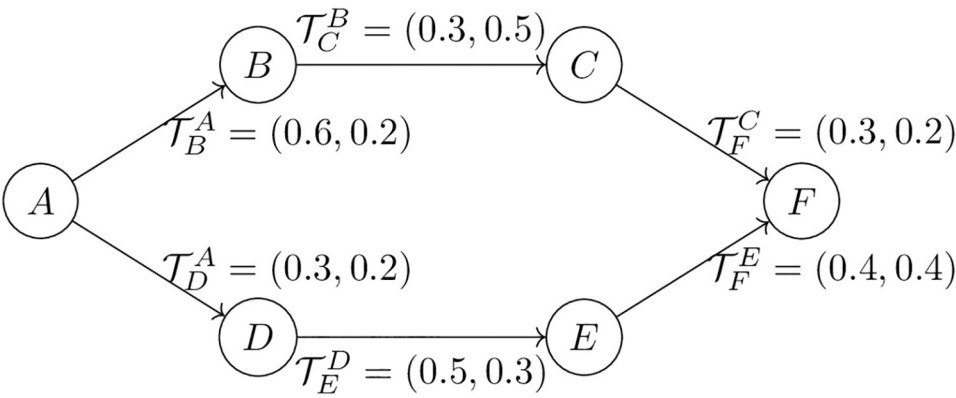

**Fig 3. Sample trust paths between nodes** *A* **and** *F* **in a CDN, where** $\mathcal{T}_X^Y$ **represents the belief and uncertainty pair** $(\mathcal{B}_X^Y, \mathcal{U}_X^Y)$ **of Y toward X.**

- Considering *TP-Discount* for propagation:

$$
\begin{aligned}
\mathcal{B}_F^{A:B:C} &= (\mathcal{B}_B^A \times \mathcal{B}_C^B \times \mathcal{B}_F^C) = 0.054 \\
\mathcal{U}_F^{A:B:C} &= 1 - \mathcal{B}_C^{A:B}(1 - \mathcal{U}_F^C) = 0.856 \\
\mathcal{T}_F^{A:B:C} &= (\mathcal{B}_F^{A:B:C}, \, \mathcal{U}_F^{A:B:C}) = (0.054, 0.856) \\
\mathcal{B}_F^{A:D:E} &= (\mathcal{B}_D^A \times \mathcal{B}_E^D \times \mathcal{B}_F^E) = 0.06 \\
\mathcal{U}_F^{A:D:E} &= 1 - \mathcal{B}_E^{A:D}(1 - \mathcal{U}_F^E) = 0.91 \\
\mathcal{T}_F^{A:D:E} &= (\mathcal{B}_F^{A:D:E}, \, \mathcal{U}_F^{A:D:E}) = (0.06, 0.91)
\end{aligned}
$$

- Considering *AP-Mean* for aggregation, assuming *TP-Discount* was used for propagation:

$$
\begin{aligned}
\mathcal{B}_F^A &= \frac{1}{2}(\mathcal{B}_F^{A:B:C} + \mathcal{B}_F^{A:D:E}) = 0.057 \\
\mathcal{U}_F^A &= \frac{1}{2}\sqrt{(\mathcal{U}_F^{A:B:C})^2 + (\mathcal{U}_F^{A:D:E})^2} = 0.625 \\
\mathcal{T}_F^A &= (\mathcal{B}_F^A, \, \mathcal{U}_F^A) = (0.057, \, 0.625)
\end{aligned}
$$

- Considering *AP-maximum* for aggregation, assuming *TP-Minimum* was used for propagation:

$$
\begin{aligned}
\mathcal{B}_F^A &= \max(\mathcal{B}_F^{A:B:C}, \, \mathcal{B}_F^{A:D:E}) = 0.3 \\
\mathcal{U}_F^A &= \min(\mathcal{U}_F^{A:B:C}, \, \mathcal{U}_F^{A:D:E}) = 0.2 \\
\mathcal{T}_F^A &= (\mathcal{B}_F^A, \, \mathcal{U}_F^A) = (0.3, \, 0.2)
\end{aligned}
$$

## 4 Evaluation design

We answer our research questions via an empirical study involving a large-scale CDN consisting of 24,315 developers spanning 179 GitHub repositories.

- To answer $RQ_1$, we investigate the accuracy of our direct and indirect trust computation approaches with respect to pre-labeled trust values.

- To answer $RQ_2$, we investigate whether a trustworthy contributor's pull request to a project is more likely to be accepted than a pull request from a less trustworthy contributor. We also investigate the practical utility of our approach by validating a predictive model that can assist an evaluator in making pull request decisions.

### 4.1 Data preparation

We select 179 Python-related GitHub projects for our analysis. These 179 projects were drawn from the sample of 918 projects created by Vasilescu et al. [19]. Each of these 918 projects had at least 200 pull requests (by 11 October 2014) in GHTorrent and used continuous integration. We select all Python projects from this sample.

All 179 projects we select use the pull request (PR) model, which is useful for two reasons. First, pull requests are a mechanism for developers to interact (via comments), which provides

evidence to estimate direct trust between developers. Second, each pull request has a clear outcome (accepted or rejected), which provides us an opportunity to compare trust between developers involved in accepted and rejected pull requests.

We use GitHub API, complying with GitHub's API terms [37], to collect all closed pull requests and associated comments for the selected projects. For each project, we collect all pull requests (since project start date) until the date of crawling (24 November, 2017). We preprocess the data as follows.

- We remove pull requests that do not contain any comments, contain comments only from the pull request generator, or contain no comment referring to the pull request generator. We treat a comment as referring to the pull request generator, if it contains "@username," where username is that of the pull request generator.

- We remove auto generated comments, specifically, those about code coverage. The body of such comments start with "[![Coverage Status]." We remove a pull request if it only contains auto generated comments.

- We remove URLs and code snippets from each comment.

- We replace frequent abbreviations with their full forms. We manually compiled a list of abbreviations that occurred in our annotated dataset. The list includes the following five abbreviations (which were expanded to their full forms as shown): (1) TBH (to be honest); (2) LGTM (looks good to me); (3) R+ (reviewed); (4) WC (welcome); and (5) BTW (by the way).

- We observed that many comments contain meaningful emojis, which are useful in determining a comment's opinion type. We replace each emoji with the corresponding description provided by Emojipedia.

We partition the data as $D_{train}$ and $D_{test}$. $D_{train}$, the training set, contains all closed pull requests until six months before data crawling (i.e., until 24 May, 2017); $D_{test}$, the test set, contains all closed pull requests in the last six months (i.e., 24 May, 2017 to 24 November, 2017). Table 3 shows the distribution of the pull requests in these datasets.

## 4.2 Experiments for RQ$_1$

First, we evaluate the opinion prediction techniques we developed for estimating direct trust based on pull request interactions. Second, we evaluate our approaches for trust propagation and aggregation based on a CDN constructed from $D_{train}$.

**4.2.1 Direct trust.** We employ our expert-annotated dataset (Section 3.2.1), consisting of 400 pull requests with 616 interactions between pull request evaluators and generators, to evaluate the opinion prediction models. We split the annotated data into training (70%) and test (30%) sets. That is, we train the regression and classification models on 70% interactions and test them on the remaining 30% interactions. We train these models using the default parameter values specified in their ScikitLearn implementations (Section 3.2.3). We measure the

**Table 3. The distribution of pull requests in the training and test datasets.**

| Dataset | Total PR | PR Status | |
|---|---|---|---|
| | | Accepted | Rejected |
| $D_{train}$ | 167, 780 | 128, 316 | 39, 464 |
| $D_{test}$ | 13, 765 | 10, 838 | 2, 927 |

**Table 4. The distribution of opinion labels in the expert-annotated PR interactions dataset used for opinion prediction.**

| Opinion label | Opinion value | Number of interactions | |
|---|---|---|---|
| | | Training | Test |
| Strongly positive | 5 | 16 | 7 |
| Weakly positive | 4 | 177 | 74 |
| Neutral | 3 | 134 | 60 |
| Weakly negative | 2 | 89 | 37 |
| Strongly negative | 1 | 15 | 7 |

model accuracy via mean absolute error (MAE), the mean of absolute differences in the predicted and expert-annotated opinion values in the test set.

We repeat each opinion prediction experiment 30 times, each time generating the training and test sets, randomly, but making sure that the 7:3 split is preserved for each opinion label. Table 4 shows the distribution of opinion labels in the training and test sets (where the numbers of interactions are the mean values from 30 repetitions).

In addition to the experiments based on the random split of the dataset as described above, we also performs an experiment that splits the dataset based on time. Our objective with this experiment is to evaluate whether opinions on earlier pull requests can predict the opinions for later pull requests. To do so, first, we sort the pull requests chronologically (i.e., according to the time at which the pull requests were created). Then, we select the interactions in the first 70% of the pull requests as the training set and the remaining interactions as the test set. Note that we do not repeat this experiment because only one 7:3 split is possible based on time.

Finally, we compare our regression and classification models with two baselines.

- A *Random Classifier* randomly assigns one of the five opinion values to each interaction in the test set.

- A *Majority Classifier* always assigns the majority (Table 4) opinion label, 4, to each interaction in the test set.

We employ the Kruskal-Wallis test [38] (a nonparametric extension of ANOVA for more than two samples) at the 5% significance level to compare the best performing regression and classification models with the two baselines. If the Kruskal-Wallis test rejects ($p < 0.05$) the null hypothesis that all samples compared come from the same distribution, we perform post hoc analysis to compare pairs of samples. To deal with multiplicity, we employ Dunn's multiple comparison test [39] with the Holm-Bonferroni correction [40] (a variant of Bonferroni adjustment, but universally stronger). Also, we measure the effect sizes (the amount of difference) via Cliff's Delta [41].

**4.2.2 Indirect trust.** To evaluate our approach for indirect trust computation, we construct a CDN employing data from $D_{train}$. We randomly select 1% of edges in the CDN as ground-truth edges. Specifically, we treat the direct trust estimated for these edges as the ground truth. We employ best performing opinion prediction techniques from the previous set of experiments to compute the direct trust of the ground truth edges.

Next, we remove the ground-truth edges from the CDN, and compute indirect trust between the nodes corresponding to these removed edges. We measure MAE as the mean of absolute differences between estimated indirect trust values and ground truth (direct) trust values. We measure MAE for different combinations of path length cutoff, propagation strategy, and aggregation strategy. We incorporate the best performing combination of direct and indirect trust estimation approaches in the $RQ_2$ experiments.

### 4.3 Experiments for RQ$_2$

We perform two experiments to answer RQ$_2$. The first experiment investigates the relationship between the computed trust values and pull request evaluation results. The second experiment builds a predictive model and evaluates its accuracy.

**4.3.1 Trust and pull request outcomes.** In the first experiment, we construct a CDN from D$_{train}$ and estimate trust between all pairs of developers in it. Then, we compute a trust value for each pull request in D$_{test}$. We compute a pull request's trust value as the mean of trust values from each member of the project (to which the pull request was made) to the pull request generator.

For a pull request in D$_{test}$, its generator may not be in D$_{train}$ because the pull request generator had not submitted any pull requests six months ago. We exclude such pull request because, we cannot estimate trust for those. After this exclusion, we were left with 1805 (out of 2927) rejected pull requests in D$_{test}$. We selected an equal number of accepted pull request from D$_{test}$, making sure that the selected pull request generators are in the CDN constructed from D$_{train}$.

We measure the difference in trust value between the 1805 rejected and 1805 accepted pull requests selected as described above via the Wilcoxon's ranksum test [38] at the 5% significance level. Also, we measure the effect sizes via Cliff's Delta [41].

**4.3.2 Predicting pull request acceptance.** In the second experiment, we develop a predictive model (a classifier) that recommends whether to accept or reject a pull request. We develop three model variants that differ in the features they employ for classification.

- The *PR-History* model is based on the historical performance of the pull request generator. It employs the number accepted and rejected pull requests by the pull request generator as two features. This model serves as a simple baseline.

- The *Trust*-based model employs a pull request's trust values (belief, disbelief, and uncertainty), computed as described in the first experiment, as its three features.

- The *Hybrid* model employs features of both *PR-History* and *Trust*-based models.

In order to train the *Trust*-based model, we must be able to compute trust between developers associated with a pull request. Thus, first, we construct a CDN corresponding to D$_{train}$ and compute trust between all pairs of developers in that CDN. Next, we randomly select 70% of pull requests in D$_{test}$ for training and 30% for testing the three model variants. For each pull request in D$_{test}$, we compute a trust value by propagating trust from members of the project (to which the pull request was submitted) to the pull request generator and taking a mean of the propagated values.

We repeat the experiment above 30 times, comparing the predictive performance of the three models as well as the added value of trust-based features. We measure predictive performances of the models via the standard classification evaluation metrics of precision, recall, and F$_1$ score. Since we repeat the experiment 30 times, we also compare the difference in performance between pairs of model variants via Dunn's multiple comparison test (for significance) and Cliff's delta (for effect size).

As Table 3 shows, our dataset, D$_{test}$, is imbalanced (where the class of accepted pull requests is considerably larger than the class of rejected pull requests). We balance the dataset by oversampling the minority class such that the final training and test sets each have an equal number of accepted and rejected pull requests.

## 5 Results and discussion

We report and discuss the results for the two research questions.

## 5.1 RQ₁: Accuracy of trust estimation

**5.1.1 Direct trust.** Fig 4 shows the MAE values of the five regression techniques we employ for direct trust estimation. Recall that these techniques make predictions in the [1, 5] range (Section 3.2). Each violin plot is based on the MAEs from 30 repetitions of the experiment. Among these technique, XGBoost and AdaBoost regressors, which make predictions based on an ensemble of trees, yield lower MAEs than the other techniques.

Fig 5 shows the MAE values of the five classification techniques we employ for direct trust estimation. Recall that these techniques make predictions from the {1, 2, 3, 4, 5} set (Section 3.2). Each violin plot is based on the MAEs from 30 repetitions of the experiment. Among these technique, the XGBoost classifier yields lowest MAE.

Table 5 shows the MAE values of the five regression and five classification techniques computed via a time-based partition of the dataset. AdaBoost yields the lowest MAE Among the regression techniques and XGBoost yields the lowest MAE among classifiers.

From the analysis above (Figs 4 and 5, and Table 5), we pick XGBoost regression, AdaBoost regression, and XGBoost classification as the best performing techniques for opinion prediction. Fig 6 compares the MAE values of our three best performing opinion prediction techniques and two baselines (random and majority-class classifiers). Based on the $p$ value from the Kruskal-Wallis test, we reject the null hypothesis that the MAEs of these techniques come from the same distribution.

We perform a pair-wise comparisons among the XGBoost regression and classification techniques, and the two baseline techniques via Dunn's multiple comparison test. Table 6

**Fig 4. Violin plots comparing the MAEs of the five regression models we employ for opinion prediction.**

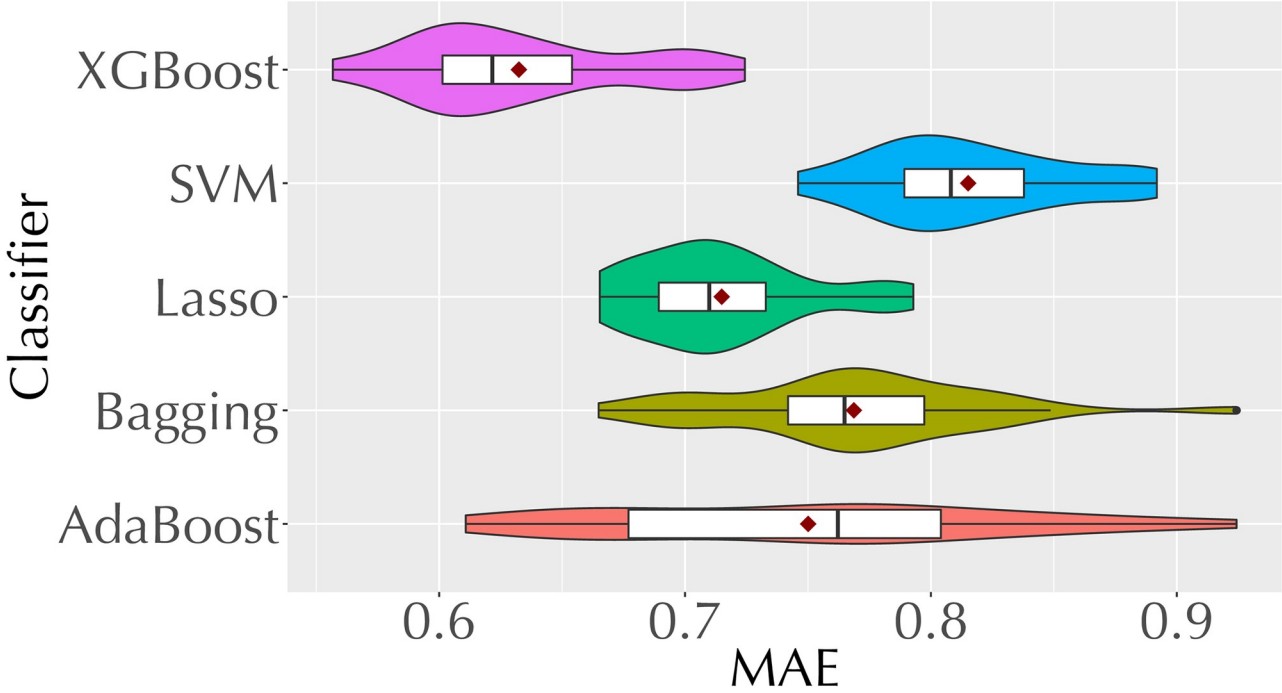

**Fig 5. Violin plots comparing the MAEs of the five classification models we employ for opinion prediction.**

shows the resulting *p* values adjust according to Holm-Bonferroni correction. Further, Table 7 shows Cliff's delta effect sizes of the differences between MAEs of different pairs of techniques.

Overall, we observe that each of the three opinion prediction models we select outperforms the baseline models with a large effect size. Among the three selected models, the XGBoost classifier yields the lowest MAE. Yet, we experiment with all three techniques in the following indirect trust estimation experiments.

**5.1.2 Indirect trust.** We experiment with three path length cutoffs, two propagation strategies and three aggregation strategies for indirect trust estimation in conjunction with the three opinion prediction techniques we selected above. Table 8 shows MAE values for each of the 12 (2×2×3) combinations of the three indirect trust estimation factors (path length, propagation, and aggregation). We make two key observations from Table 8.

**Table 5. MAEs of the five regression and five classification techniques for opinion prediction computed via a time-based analysis.**

| Technique | MAE | |
|---|---|---|
| | **Regression** | **Classification** |
| XGBoost | 0.748 | **0.665** |
| SVM | 0.785 | 0.827 |
| Lasso | 0.752 | 0.752 |
| Bagging | 0.767 | 0.800 |
| AdaBoost | **0.719** | 0.870 |

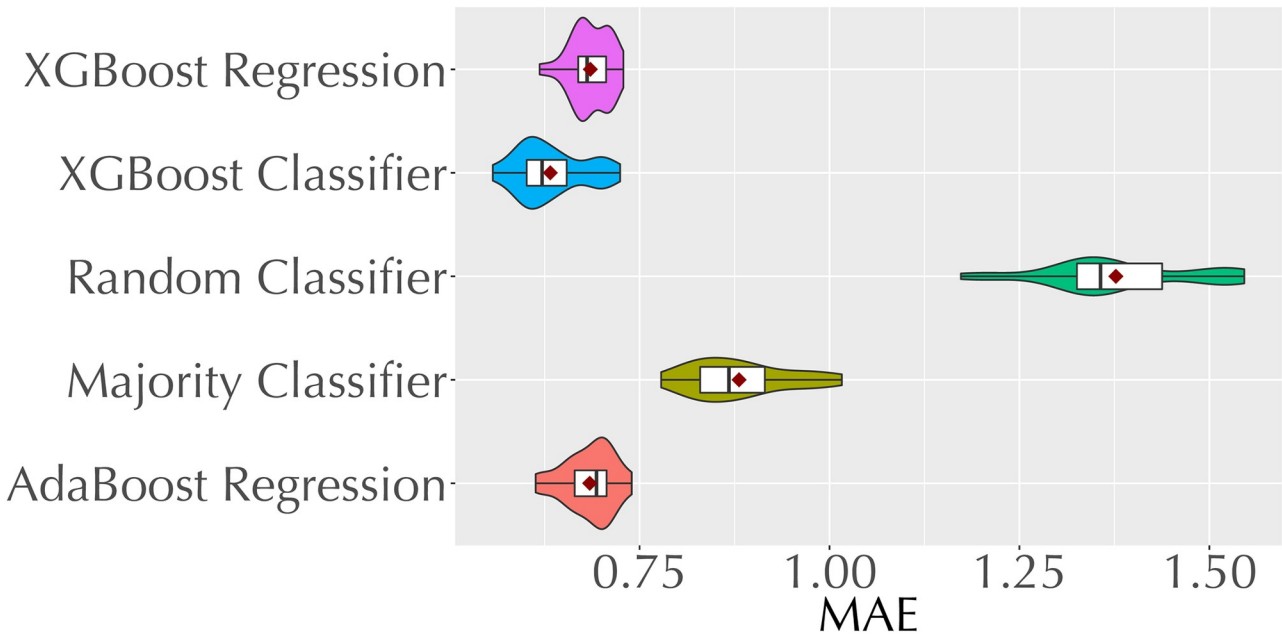

**Fig 6. Violin plots comparing the MAEs of our best performing regression and classification techniques, and two baselines models.**

- First, XGBoost regression yields lower MAEs than the other two opinion prediction techniques for most combinations of indirect trust estimation factors.

- Second, in conjunction with XGBoost regression for opinion prediction, the combination of (1) length cutoff 3, *TP-Discount* propagation and *AP-Maximum* aggregation yields lowest MAE for belief; and (2) length cutoff 2, *TP-Discount* propagation and *AP-Consensus* aggregation yields lowest MAE for uncertainty.

Overall, considering the MAEs of both belief and uncertainty, we employ the combination of XGBoost regression, path length cutoff 3, *TP-Discount* propagation, and *AP-Maximum* aggregation as the best performing combination for (direct and indirect) trust estimation in our dataset. We employ this combination in the $RQ_2$ experiments.

### 5.2 RQ$_2$: Exploiting trust in pull request evaluation

Recall that we conduct two experiments to answer $RQ_2$.

**Table 6. Pair-wise comparisons, showing the Holm-Bonferroni *p*-values, between our best performing classification (C) and regression (R) techniques, and the baselines classification (C) models.**

| Technique | Holm-Bonferroni *p*-value | | | |
|---|---|---|---|---|
| | **AdaBoost (R)** | **Majority (C)** | **Random (C)** | **XGBoost (C)** |
| Majority (C) | <0.0001 | | | |
| Random (C) | <0.0001 | 0.0299 | | |
| XGBoost (C) | 0.0221 | <0.0001 | <0.0001 | |
| XGBoost (R) | 0.9502 | <0.0001 | <0.0001 | 0.0277 |

**Table 7. Pair-wise comparisons, showing the Cliff's delta effect sizes, between our best performing classification (C) and regression (R) techniques, and the baselines classification (C) models.**

| Technique | Cliff's Delta | | | |
|---|---|---|---|---|
| | **AdaBoost (R)** | **Majority (C)** | **Random (C)** | **XGBoost (C)** |
| Majority (C) | −1 (large) | | | |
| Random (C) | −1 (large) | −1 (large) | | |
| XGBoost (C) | 0.62 (large) | 1 (large) | 1 (large) | |
| XGBoost (R) | −0.01 (negligible) | 1 (large) | 1 (large) | −0.66 (large) |

- In the first experiment (Section 5.2.1), we investigate the relationship between trust and pull request outcomes (acceptance or rejection). We analyze this relationship, considering trust dimensions individually as well as jointly.

- In the second experiment (Section 5.2.2), we investigate the predictive performance of the classifiers we build for assisting in pull request evaluation. We compare the three classifier variants (differing in features they employ).

**5.2.1 Trust and pull request outcomes.** Fig 7 shows a comparison of the trust values between accepted and rejected pull requests. We find that there is a significant difference in each trust dimension between accepted a rejected pull requests. Specifically, accepted pull requests are associated with higher belief, lower disbelief, and lower uncertainty values, with small, negligible, and small effect sizes, respectively, compared to rejected pull requests. This finding establishes that *the inferred trust and pull request outcomes are related.*

Further, the effect sizes in Fig 7 provide additional insights.

**Table 8. MAEs of indirect trust estimation (computed on ground truth edges from $D_{train}$) for different combinations of path length cutoff, and propagation, aggregation, and opinion prediction techniques.**

| Opinion Prediction | Transitivity | Aggregation | MAE | | | | | |
|---|---|---|---|---|---|---|---|---|
| | | | **Path Length = 2** | | **Path Length = 3** | | **Path Length = 4** | |
| | | | $\mathcal{B}$ | $\mathcal{U}$ | $\mathcal{B}$ | $\mathcal{U}$ | $\mathcal{B}$ | $\mathcal{U}$ |
| XGBoost Regression | *TP-Minimum* | *AP-Mean* | 0.089 | 0.189 | 0.089 | 0.274 | 0.091 | 0.273 |
| | *TP-Minimum* | *AP-Maximum* | 0.092 | 0.135 | 0.093 | 0.137 | 0.094 | 0.137 |
| | *TP-Minimum* | *AP-Consensus* | 0.095 | 0.156 | 0.122 | 0.359 | 0.117 | 0.508 |
| | *TP-Discount* | *AP-Mean* | 0.090 | 0.167 | 0.094 | 0.245 | 0.096 | 0.250 |
| | ***TP-Discount*** | ***AP-Maximum*** | 0.090 | 0.125 | **0.080** | **0.127** | 0.084 | 0.128 |
| | *TP-Discount* | *AP-Consensus* | 0.090 | 0.120 | 0.101 | 0.143 | 0.119 | 0.174 |
| AdaBoost Regression | *TP-Minimum* | *AP-Mean* | 0.151 | 0.255 | 0.175 | 0.265 | 0.230 | 0.270 |
| | *TP-Minimum* | *AP-Maximum* | 0.199 | 0.231 | 0.222 | 0.245 | 0.250 | 0.250 |
| | *TP-Minimum* | *AP-Consensus* | 0.215 | 0.363 | 0.242 | 0.546 | 0.290 | 0.530 |
| | *TP-Discount* | *AP-Mean* | 0.185 | 0.243 | 0.230 | 0.255 | 0.300 | 0.260 |
| | ***TP-Discount*** | ***AP-Maximum*** | 0.130 | 0.168 | **0.133** | **0.170** | 0.170 | 0.200 |
| | *TP-Discount* | *AP-Consensus* | 0.185 | 0.229 | 0.299 | 0.425 | 0.430 | 0.490 |
| XGBoost Classification | *TP-Minimum* | *AP-Mean* | 0.180 | 0.260 | 0.210 | 0.270 | 0.230 | 0.270 |
| | *TP-Minimum* | *AP-Maximum* | 0.220 | 0.240 | 0.250 | 0.250 | 0.250 | 0.250 |
| | *TP-Minimum* | *AP-Consensus* | 0.260 | 0.370 | 0.300 | 0.510 | 0.290 | 0.530 |
| | *TP-Discount* | *AP-Mean* | 0.210 | 0.250 | 0.270 | 0.260 | 0.300 | 0.260 |
| | ***TP-Discount*** | ***AP-Maximum*** | 0.170 | 0.200 | **0.170** | **0.200** | 0.170 | 0.200 |
| | *TP-Discount* | *AP-Consensus* | 0.230 | 0.280 | 0.370 | 0.440 | 0.430 | 0.490 |

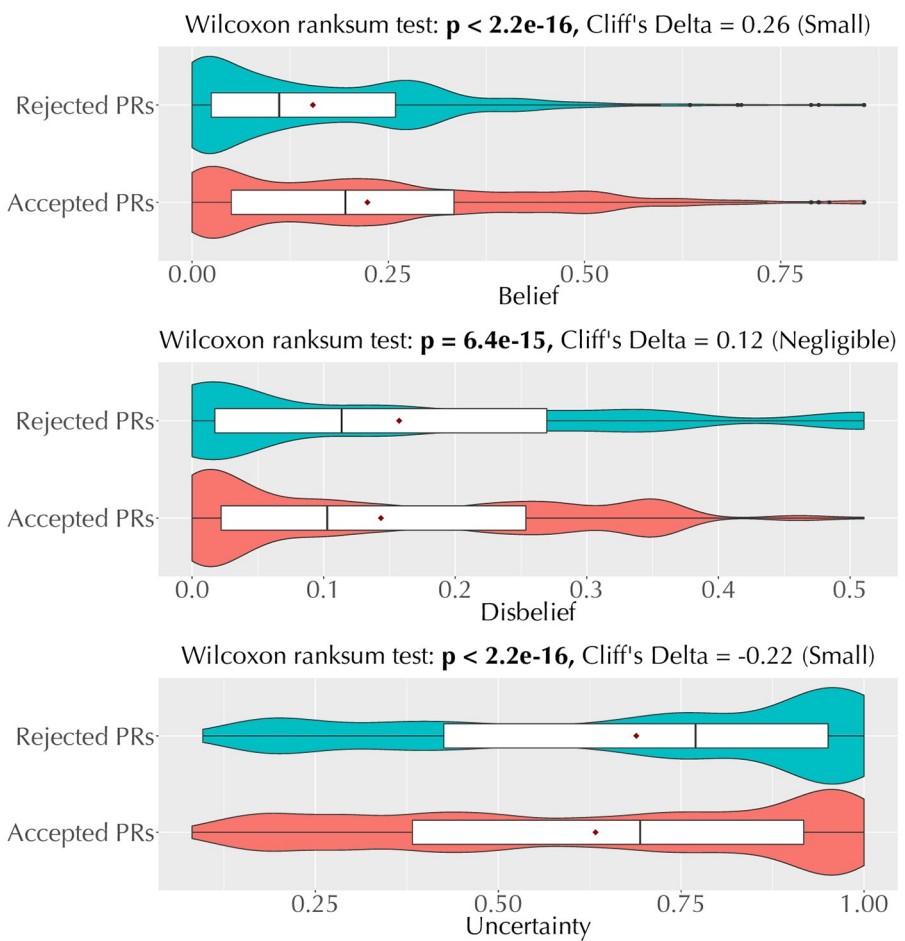

**Fig 7. Violin plots comparing individual trust dimensions (belief, disbelief, and uncertainty) between accepted and rejected pull requests.**

- The small (but non-negligible) effect sizes for belief and uncertainty suggest that a pull request from a trustworthy (high belief) contributor is more likely to be accepted than rejected. The small effect size is not surprising since trust is likely to be one among several factors that may influence pull request outcomes.

- The negligible effect for disbelief is interesting. It suggests that the higher likelihood of a pull request from a distrustworthy (high disbelief) contributor being rejected instead of accepted is negligible.

*Joint Analysis of Trust Dimensions.* Fig 7 suggests that the dimensions of the inferred trust and pull request outcomes are related. However, interpreting a trust relationship as trustworthy, distrustworthy, or lacking trust requires a joint interpretation of belief, disbelief, and uncertainty (Table 1). For instance, a trustworthy relationship has high belief, low disbelief, and low uncertainty. Fig 8 shows a joint comparison of trust dimensions between accepted and rejected pull requests.

The blue data points in Fig 8 are in the trustworthy regions. It is evident that there are more data points in the trusted region for accepted pull request than for rejected pull requests. Table 9 quantifies this difference by showing the exact number of data points in each region of trust, defined as follows, for a given uncertainty threshold ($u$).

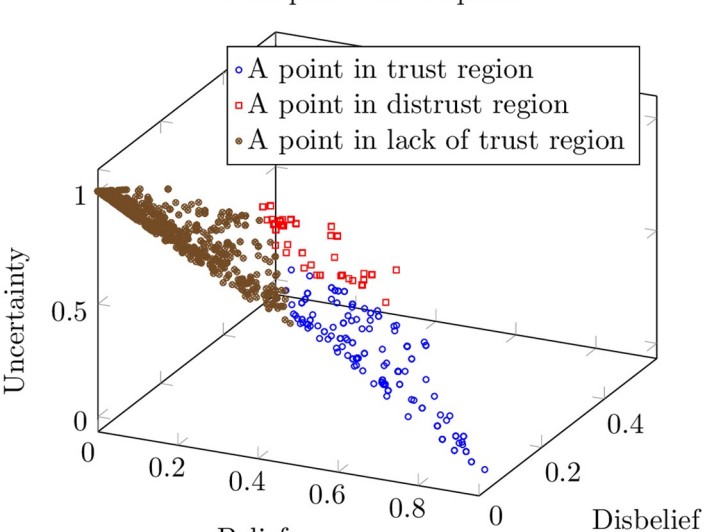

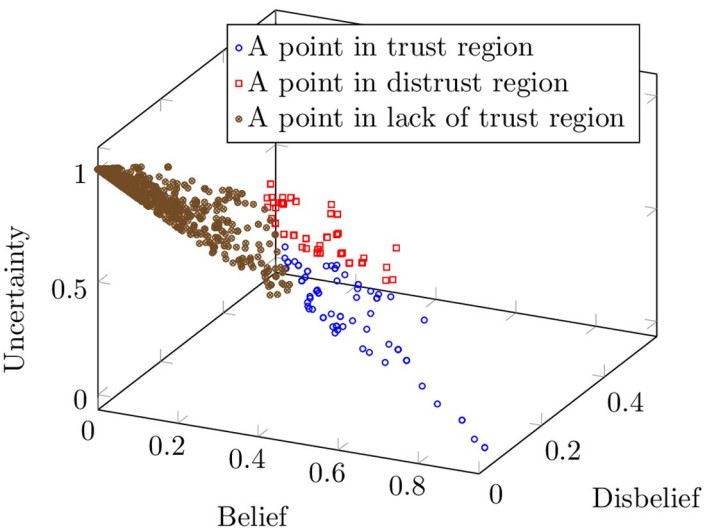

**Fig 8. 3D plots showing a joint comparison of trust dimensions (belief, disbelief, and uncertainty) between accepted and rejected pull requests.** The pull requests in the regions of trust, distrust, and lack of trust are shown as data points in blue, red, and brown, respectively.

- In the region of *trust*: $\mathcal{B} \geq \mathcal{D}$ and $\mathcal{U} < u$.

- In the region of *distrust*: $\mathcal{D} > \mathcal{B}$ and $\mathcal{U} < u$.

- In the *lack of trust* region: $\mathcal{U} \geq u$.

As Table 9 shows, based on the $\chi^2$ test of homogeneity, we find that the frequency counts of accepted and rejected pull requests are significantly different between the three regions of trust. Specifically, we observe that:

**Table 9. The number of accepted and rejected pull requests in each of the three trust regions (trust, distrust, and lack of trust).**

| Uncertainty threshold | PR status | Number of points | | | ($\chi^2$ test) |
|---|---|---|---|---|---|
| | | **Trust** | **Distrust** | **Lack of trust** | **p-value** |
| $u = 0.25$ | Accepted | 72 | 7 | 1726 | $1.3e^{-30}$ |
| | Rejected | 20 | 5 | 1780 | |
| $u = 0.50$ | Accepted | 193 | 142 | 1470 | $4.4e^{-46}$ |
| | Rejected | 78 | 95 | 1632 | |
| $u = 0.75$ | Accepted | 506 | 261 | 1038 | $1.5e^{-105}$ |
| | Rejected | 232 | 162 | 1411 | |

- There are considerably more accepted pull requests than rejected pull requests in the trusted region. This confirms our earlier observation (based on Fig 7) that pull requests from trusted contributors are more likely to be accepted than rejected.

- Surprisingly, there are more accepted than rejected pull requests in the region of distrust, too. However, the difference is not as significant as it is for the region of trust. Further, the difference gets narrower as we reduce the uncertainty threshold (i.e., as we get more certain about our belief or disbelief).

- Finally, there are more rejected than accepted pull requests in the region lacking trust. Further, similar to distrust, the difference is not considerably large and it narrows as we reduce the uncertainty threshold used for defining the regions.

In essence, a pull request from a trusted developer is more likely to be accepted to a project than rejected. However, a pull request from a developer who is distrusted or lacks trust does not have a higher likelihood of getting rejected than accepted.

**5.2.2 Predicting pull request acceptance.** Our analysis above (Section 5.2.1) establishes that pull request outcomes and trust between the associated developers are related. Next, we seek to exploit this relationship in a model that can predict whether a pull request is likely to be accepted or rejected.

We train a decision tree [42], a well-known classification technique, for each model variant. Fig 9 compares the predictive performance of the three model variants and Table 10 shows the confusion matrix for each model variant. Each confusion matrix (2×2) shows the number of true positives (top left cell), false positives (bottom right cell), true negatives (bottom right cell), and false negatives (top right cell). Note that we report these numbers as the mean of 30 repetitions.

We also perform a Dunn's multiple comparisons test among the three model variants. Table 11 shows the Holm-Bonferroni adjusted $p$ values resulting from the pair-wise comparisons. Similarly, Table 12 shows the Cliff's Delta effect sizes from the pair-wise comparisons. From this analysis, we find that (1) *Trust*-based model yields a high $F_1$ score and performs significantly better than the *PR History* based model. (2) The *Hybrid* model, which employs both *PR History* and *Trust* based features, improves the performance further, albeit, by a small margin.

Our findings suggest that the *Trust*-based and *Hybrid* predictive models can recommend whether to accept or reject a pull request with a high accuracy. Although we do not expect pull request evaluations to be fully automated, recommendations from our model can assist pull request evaluators. For example:

- Consider that a project has a large number of open pull requests, but only a few developers available to evaluate those pull requests. Then, project evaluators can employ our predictive

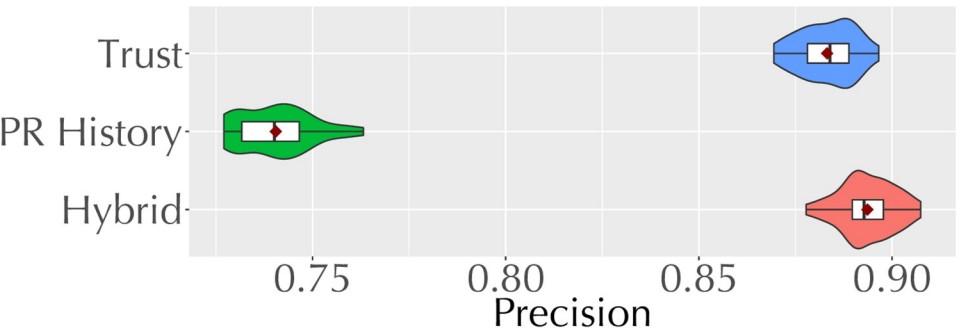

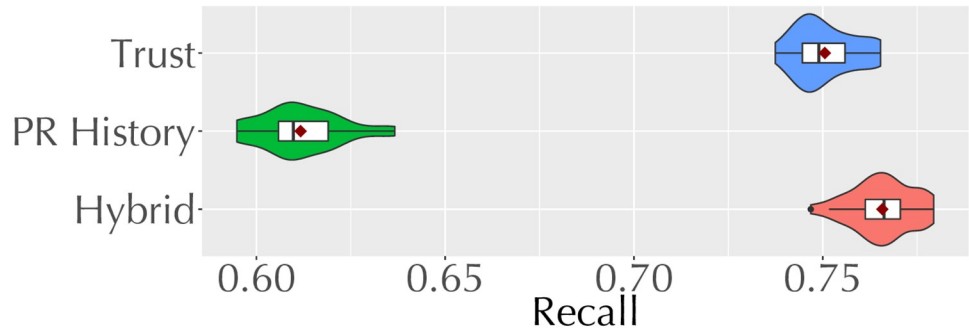

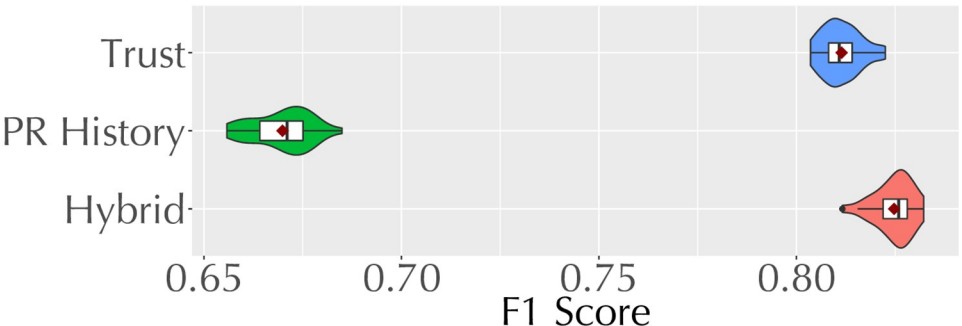

**Fig 9. Violin plots comparing the precision, recall, and F$_1$ scores of the three model variants for predicting pull request acceptance.**

model to prioritize which pull requests to evaluate first (e.g., they may evaluate pull requests recommended as accept by our model, first).

- Conider that an evaluator wants to accept a pull request but our model recommends rejection or vice versa. In such a case, our recommendation can serve as a warning, suggesting the evaluator to double check the contribution.

**Table 10. The confusion matrices for the three model variants.** Each confusion matrix is a 2×2 matrix with the same background color.

| Predicted<br>Actual | PR History | | Trust | | Hybrid | |
|---|---|---|---|---|---|---|
| | Accept | Reject | Accept | Reject | Accept | Reject |
| Accepted | 1959.43 | 1243.83 | 2401.20 | 798.13 | 2461.20 | 752.77 |
| Rejected | 686.87 | 2520.87 | 317.63 | 2894.03 | 293.13 | 2903.90 |

**Table 11. Pair-wise comparisons of the predictive performance, showing the Holm-Bonferroni _p_-values, of the three model variants we develop for predicting pull request acceptance.**

| Variant | Holm-Bonferroni _p_-value | | | | | |
|---|---|---|---|---|---|---|
| | Precision | | Recall | | F$_1$ score | |
| | _Hybrid_ | _PR History_ | _Hybrid_ | _PR History_ | _Hybrid_ | _PR History_ |
| _PR History_ | <0.0001 | | <0.0001 | | <0.0001 | |
| _Trust_ | 0.0013 | <0.0001 | 0.0003 | <0.0001 | <0.0001 | <0.0001 |

## 5.3 Threats to validity

We identify four threats to the validity of our findings.

_Lack of Trust_. As Fig 8 and Table 9 suggest, _lack of trust_ is the largest of the three trust regions in our dataset since the uncertainty values we compute are typically on the higher end of the spectrum. We attribute the high uncertainty to the limited evidence available for trust computation. First, we compute direct trust via a regression model trained and tested on a small dataset (702 comments spanning 400 pull requests). Thus, for most developer pairs, we estimate direct trust based on evidence from a few interactions (one interaction in many cases), which yields high uncertainty. The uncertainty increases further as we propagate trust during indirect trust computation.

The pull request distributions across the trust regions are likely to change as more evidence is incorporated into trust computation (e.g., by increasing annotated examples). In that case, we conjecture that the number of(1) accepted pull requests will increase in the trusted region, (2) rejected pull requests will increase in the distrusted region, and (3) accepted and rejected pull requests will be similar in the region lacking trust.

_Word Embedding_. During feature extraction for opinion prediction, we take the mean of the word vectors of a comment to get the vector representation of the comment. Although a simple word averaging technique for sentence representation has been successful in some existing applications, e.g., [43, 44], this technique can be suboptimal.

_Predictive Models_. We train and test the predictive models for pull request evaluation on D$_{test}$, which is a small dataset (capturing pull requests from only six months). Further, we train these models on decision trees, a simple classification technique. Thus, the classification accuracy metrics we report in Fig 9 are not indicative of the highest performance a predictive model can achieve in evaluating pull requests. We conjecture that more sophisticated classification techniques (e.g., deep learning techniques) trained on larger datasets can perform better than the decision tree models we employ. That said, our objective was not to find the best classification technique but to demonstrate the practical utility of a trust-based predictive model, which we do via a simple classification technique.

_Generalizability_. First, we analyze opinion prediction techniques via MAEs averaged across repositories. However, the standards of opinions can vary across repositories. For example, different repositories may have different opinions on what is an acceptable contribution. Our

**Table 12. Pair-wise comparisons of the predictive performance, showing the Cliff's delta effect sizes, of the three model variants we develop for predicting pull request acceptance.**

| Variant | Cliff's Delta | | | | | |
|---|---|---|---|---|---|---|
| | Precision | | Recall | | F$_1$ score | |
| | _Hybrid_ | _PR History_ | _Hybrid_ | _PR History_ | _Hybrid_ | _PR History_ |
| _PR History_ | 1 (large) | | 1 (large) | | 1 (large) | |
| _Trust_ | 1 (large) | −1 (large) | 0.72 (large) | −1 (large) | 0.80 (large) | −1 (large) |

analyses do not provide sufficient evidence on whether or not our opinion prediction techniques generalize across repositories with different standards of opinions. Performing such an analysis requires a dataset with several annotated interactions from each repository (which is not the case with our current dataset).

Second, we construct a CDN, including Python projects on GitHub, employing pull request comments as developer interactions. The generalizability of our findings beyond this setting (e.g., for a community of Java developers communicating via a mailing list) remains to be verified. We defer such efforts to future work.

## 6 Related works

We briefly describe related works on trust in online collaboration and computational approaches for estimating trust.

### 6.1 Trust in OSS projects

Trust is essential to successful teamwork in OSS projects [3, 45, 46]. Trust plays a vital role in maintaining high cohesion between team members, and thus, promoting cooperation [47]. A trusted team can attract new developers [48]. Thus, an OSS project's sustainability depends on the trustworthiness of the project's developers [49].

OSS teams are distributed and virtual, exhibiting characteristics such as lack of face-to-face interactions and low awareness of others' activity. Such characteristics make trust building a big challenge. Researchers have studied trust in OSS projects from multiple perspectives. Jarvenpaa et al. [7, 10] propose contextualized theories on how trust is developed in virtual teams. Wang et al. [9, 50, 51], describe the emergence, diffusion, and other dynamics of trust in networked OSS teams, while Trainer and Redmiles [11] discuss how such dynamics could be supported with computing tools.

Zolin et al. [52] study how trust impacts team process in distributed software development. Steward and Gosain [53] claim that OSS projects involving more trusted developers are more likely to succeed compared to those involving less trusted developers. At the individual level, researches have found the importance of trust in the many decision-making scenarios in OSS [54–56]. For instance, Gousios et al. [18] show that trust between project member and contributor is an influential factor granting the contribution to the OSS project. On a similar line, Sinha et al. [17] state that trustworthiness is a key factor in letting an unknown developer contribute to a project. Calefato and Lanubile [57] find that a developer with a high propensity to trust is more likely to accept the contributions from external contributors.

Although there is an increasing emphasis on the essential role of trust in OSS development [45, 48], judging the trustworthiness of unfamiliar developers is still a significant challenge. A few tools, e.g., Theseus [11], have been developed to compute collaborators' trust. However, Theseus relies on individual interaction traces in a single project, ignoring the network characteristics. Our computational approach leverages individual interactions as well as the community-wide developer network to estimate trust, significantly broadening the scope of trust computation in an OSS community.

### 6.2 Trust inference and propagation

Trust is a widely studied topic in reputation systems [58], multiagent systems [59], social networks [60], and internet applications [61], in general. Since our contribution involves inferring trust in social networks, we describe works closely related to that.

Our approach is based on Jøsang's subjective logic [22] which defines trust in *opinion* and *evidence* spaces. We map trust values from evidence to opinion space using the mapping

function provided by Jøsang [58]. For indirect trust referral, subjective logic defines two operators: trust transitivity and cumulative fusion. The basic notion of trust transitivity is that belief is discounted by all the belief values that lie in a trust chain. That is, the transitivity operator discounts belief and increases uncertainty as the length of a chain increases. The cumulative fusion operator combines trust values from multiple chains, amplifying belief and disbelief, and reducing uncertainty [16].

A variety of techniques have been proposed in the literature for trust inference. For instance, Golbeck [21] infers trust based on ratings and referrals only from trusted nodes. Hamdi et al. [62] compute trust in an online social network based on interactions, relationship types, and interest similarity between users. Guha et al. [63] and Zhao et al. [64] use machine learning to compute indirect trust. Kafali and Yolum [65] propose reinforcement learning-based approach to measuring trust between agents. in a multiagent setting. Richters and Peixoto [66] investigate different trust transitivity techniques and analytically obtain the average best trust transitivity technique. Liu et al. [67] propose a trust transitivity model, MQCTT, that considers social relationships, recommendation roles, and preference similarity in determining transitive trust value.

Ruan et al. [36] evaluate different trust transitivity and aggregation operators. Multiplication is the first transitivity operator they define based on exiting works [21, 68]. The multiplication operator discounts trust by the values that lie on a transitive path. Their second transitivity operator, based on Sun et al. [69], captures the notion that enemy of an enemy is a friend. Their third transitivity operator is based on minimum t-norm [70], which takes a minimum of trust values associated with all edges in the chain. Similar to transitivity, Ruan et al. [36] also define different aggregation operators. Their first aggregation operator takes the mean of trust values obtained from multiple trust paths [71]. Their second operator takes a weighted, instead of simple, mean of trust values based on other works [21, 68]. Their third operator is based on the law of probability [69]. Their fourth operator is based on the notion that an evaluator chooses the path among parallel path that has the highest trust [72]. We select trust propagation operators based on their performance on our dataset.

Although there is an extensive body of research on trust, our work is unique for two reasons. First, our work is the first to apply trust computation and propagation to an OSS developer network. Second, given the variety of methods and operators available to compute trust, selecting the appropriate ones is context dependent [36]. We explore several techniques in choosing promising methods for estimating trust in OSS developer networks and empirically demonstrate the utility of those methods.

## 7 Conclusions

We propose a network-centric approach for estimating trust between developers in an OSS community. Our approach consists of three key steps: (1) constructing a community-wide network according to historical collaboration traces, (2) calculating the trust between directly connected developers, and (3) inferring the trust between indirectly connected developers in the network. To the best of our knowledge, our work is one of the first attempts at computationally estimating trust between OSS developers.

We build a dataset based on 179 Python-related GitHub projects and perform extensive analyses to answer two research questions. Our analyses demonstrate the high accuracy of the proposed approaches for direct and indirect trust estimation as well as the practical utility of the computed trust metrics. Specifically, we find that trusted developers are likely to be rewarded (i.e., their contributions to a project have a higher likelihood of being accepted than rejected by the project members). Exploiting this finding, we develop a predictive model that

can recommend whether to accept or reject a pull request based on the trustworthiness of the contributor. We find that the predictive model, though based on a simple classification technique, yields high accuracy. Such a predictive model can be employed in tasks such as recommending contributors to projects and projects to OSS developers, warning developers about untrustworthy contributions, and prioritizing pull requests for evaluation.

## Author Contributions

**Conceptualization:** Hitesh Sapkota, Pradeep K. Murukannaiah, Yi Wang.

**Data curation:** Hitesh Sapkota.

**Investigation:** Hitesh Sapkota, Pradeep K. Murukannaiah, Yi Wang.

**Methodology:** Pradeep K. Murukannaiah, Yi Wang.

**Project administration:** Pradeep K. Murukannaiah.

**Software:** Hitesh Sapkota.

**Supervision:** Pradeep K. Murukannaiah, Yi Wang.

**Validation:** Hitesh Sapkota, Pradeep K. Murukannaiah, Yi Wang.

**Visualization:** Hitesh Sapkota, Pradeep K. Murukannaiah, Yi Wang.

**Writing – original draft:** Hitesh Sapkota, Pradeep K. Murukannaiah, Yi Wang.

**Writing – review & editing:** Hitesh Sapkota, Pradeep K. Murukannaiah, Yi Wang.

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
