## [Decision Letter · Decision Letter 0]

2 Sep 2019

PONE-D-19-20119

A Network-Centric Approach for Estimating Trust between Open Source Software Developers

PLOS ONE

Dear Dr Murukannaiah,

Thank you for submitting your manuscript to PLOS ONE. After careful consideration, we feel that it has merit but does not fully meet PLOS ONE’s publication criteria as it currently stands. Therefore, we invite you to submit a revised version of the manuscript that addresses the points raised during the review process.

Please refer to the attached reviewer's comment and address them in detail in your re-submission.

We would appreciate receiving your revised manuscript by Oct 17 2019 11:59PM. To enhance the reproducibility of your results, we recommend that if applicable you deposit your laboratory protocols in protocols.io, where a protocol can be assigned its own identifier (DOI) such that it can be cited independently in the future. For instructions see: http://journals.plos.org/plosone/s/submission-guidelines#loc-laboratory-protocols

We look forward to receiving your revised manuscript.

Kind regards,

Tiago P. Peixoto

Academic Editor

PLOS ONE

Journal Requirements:

1. In your Methods section, please include additional information about your dataset and ensure that you have included a statement specifying whether the collection method complied with the terms and conditions for the website.

Reviewers' comments:

Reviewer's Responses to Questions

**Comments to the Author**

1. Is the manuscript technically sound, and do the data support the conclusions?

Reviewer #1: Partly

2. Has the statistical analysis been performed appropriately and rigorously? 

Reviewer #1: Yes

3. Have the authors made all data underlying the findings in their manuscript fully available?

Reviewer #1: Yes

4. Is the manuscript presented in an intelligible fashion and written in standard English?

Reviewer #1: Yes

5. Review Comments to the Author

Reviewer #1: Summary

-------

The authors describe a method for representing trust relationships within GitHub projects using pull requests. They develop a labeled dataset of comments indicating "trusted language" (or lack thereof), and test their methodology on this dataset. They further use this method to create a classifier for pull request acceptance or rejection, with a comparison to a simple baseline.

The authors frame their work in the context of prior work regarding trust, and motivate it in a sound manner. In addition, the paper is very well written and easy to read and follow.

There are a number of revisions I would like to see made and / or discussed by the authors before I recommend publication.

Review

-------

* In general, there are a few spots in which some conclusions are drawn that, I'd argue, are not fully supported by the data; an example is outlined below (line 428). As the primary consideration is experimental robustness and not "practical applicability", I would suggest toning back some of the stronger results statements. Most statements are acceptable and the subtlety is appreciated (e.g., lines 407-412 and lines 418-419).

* Line 43: How is a "top" Python project defined?

* Line 92-95: This description is somewhat confusing. How does each parameter take a value in [0, 1], yet three parameters add up to 1? I believe this is better explained later in the paper, but some small introduction or explanation here would make things a bit easier to follow.

* Line 209-210: To my knowledge, there is no precedent that the mean of word vectors in a document is representative of the content as a whole; only that word vectors are pair-wise close to each other in embedding space as measured by cosine similarity (and in compositions of cosine similarity). I would suggest citing a reference to such work, or explicitly stating this as a threat to validity.

* Section 3.2.3: It is unclear at what level the observations (i.e., "rows") are for the regression. Are they per-comment? Per pull request? Later (in results) this becomes clear, but stating it here may be a good idea.

* Fig. 2 / Section 3.2.3: What type of cross-validation was performed? K-fold (with what value of k)? Would results change if cross-validation was performed with respect to time? Contribution guidelines and the expected quality of accepted pull requests may change drastically over time, and may affect underlying trust relationships. Would results change if cross-validation was performed across repositories? Projects have different views of acceptable pull request quality / submission behavior, which will likely affect trust relationships.

* Line 329-331: Re (2): How are auto-generated comments detected? Re (4): What precisely constitutes "referring to the pull requester"? The usage of an @-mention directed towards the pull requester? Or some other signaling mechanism?

* Line 333: How was this replacement performed? Is a list of mappings between abbreviations and their full forms available?

* Section 4.2 / 5.1: Interpreting performance based on MAE without a clear baseline is difficult. What is the MAE in the worst case scenario? E.g., when always assigning a 5 when the true value is a 1, and analogously assigning maximum "worst possible values" to the test data. What would the MAE be if one, e.g., randomly assigned labels (uniformly from 1 to 5) to the test set? A table or figure representing the distribution of labels in the train and test set would also be helpful here. In addition, I would also appreciate a variance-based (or at least "variance informed") evaluation metric in addition to MAE.

Was classification, as opposed to regression, also considered here? I understand that the regression approach preserves distance between classes defined on a Likert (or Likert-like) scale, but I am curious as to whether the authors believe a classification approach would drastically change their results.

* Section 4.3 / 5.2.2: I very much appreciate the usage of a simple baseline model for comparison. Is there a reason that you did not include features used in prior work (such as in the cited work by Tsay et al. [14], who build a model for pull request acceptance as done in this work)?

* Section 5: As stated above, it is unclear what method of cross-validation is performed. Please clarify.

* Table 5: I assume this performance is calculated on the test set, but an explicit statement of such in the table caption would be useful.

* Line 381-388: It is stated that trust is computed in the CDN described by D_train. Is this trust value then considered "fixed" and used in the corresponding models trained on 70% of data from D_test? A bit of additional explicit description and clarification could be added here.

* Line 428: The statement that Fig 4 provides "strong evidence" for the given conclusion is a stretch; Cliff's delta is either small or negligible in all cases. Though journal guidelines dictate that research "impact" is not a considered guideline for publication, this statement of "strong evidence" is a stretch. I'd suggest removing the "strong" adjective here.

* Fig 5: This plot is hard to read (both in image quality and presentation). The addition of a contour plot, estimated 3-d surface, or some other positional indicator(s) (e.g., a line drawn from the horizontal plane vertically to each point on the vertical axis) would greatly increase readability.

* Section 5.2.1 / Table 6: What part of the data are these values drawn from? The 30% of D_test described in Section 4.3? Table 3 states that there are 13,765 pull requests in the test set, but there are 3,610 data points in each cell of Table 6.

Misc. Minor Comments

-------

* Line 197-200: "one annotator labelled 200 pull requests, and another annotator labelled an a different set of 100 pull requests"; it is not clear that the "another annotator" is the same annotator for which ICC was calculated in prior sentences.

Grammar / Typos:

* Line 132: "that adopt pull request model"

* Line 199: "... another annotator labelled an a different ..."

* Line 234: "For each comment evaluator makes on ..."

* Line 256: "i.e." should be italicized

* Line 450: "i.e." should be italicized

6. PLOS authors have the option to publish the peer review history of their article (what does this mean?). If published, this will include your full peer review and any attached files.

Reviewer #1: No

---

## [Author Response · Author response to Decision Letter 0]

1 Nov 2019

Thanks to the editor and the reviewers for very helpful comments.

We have addressed all the comments in the revision. For each comment, we identify the page number of the manuscript on which we made the pertinent changes; on the specified page of the manuscript with tracked changes, we have inserted an arrow in the margin (with the specified tag) that points approximately to the changed text.

---

## [Decision Letter · Decision Letter 1]

25 Nov 2019

A Network-Centric Approach for Estimating Trust between Open Source Software Developers

PONE-D-19-20119R1

Dear Dr. Murukannaiah,

We are pleased to inform you that your manuscript has been judged scientifically suitable for publication and will be formally accepted for publication once it complies with all outstanding technical requirements.

With kind regards,

Tiago P. Peixoto

Academic Editor

PLOS ONE

Additional Editor Comments (optional):

Reviewers' comments:

Reviewer's Responses to Questions

**Comments to the Author**

1. If the authors have adequately addressed your comments raised in a previous round of review and you feel that this manuscript is now acceptable for publication, you may indicate that here to bypass the “Comments to the Author” section, enter your conflict of interest statement in the “Confidential to Editor” section, and submit your "Accept" recommendation.

Reviewer #1: All comments have been addressed

2. Is the manuscript technically sound, and do the data support the conclusions?

Reviewer #1: Yes

3. Has the statistical analysis been performed appropriately and rigorously? 

Reviewer #1: Yes

4. Have the authors made all data underlying the findings in their manuscript fully available?

Reviewer #1: Yes

5. Is the manuscript presented in an intelligible fashion and written in standard English?

Reviewer #1: Yes

6. Review Comments to the Author

Reviewer #1: Thank you to the author for addressing all listed concerns explicitly, concisely, and concretely. Particularly, it appears extensive additional effort was expended in order to completely address my concerns (e.g., additional experiments, variance-informed comparisons, additional baselines, etc.), which is greatly appreciated.

This manuscript is now complete and all my concerns have been addressed.

7. PLOS authors have the option to publish the peer review history of their article (what does this mean?). If published, this will include your full peer review and any attached files.

Reviewer #1: No

---

## [Editor Report · Acceptance letter]

3 Dec 2019

PONE-D-19-20119R1 

A Network-Centric Approach for Estimating Trust between Open Source Software Developers 

Dear Dr. Murukannaiah:

I am pleased to inform you that your manuscript has been deemed suitable for publication in PLOS ONE. Congratulations! Your manuscript is now with our production department. 

With kind regards,

on behalf of

Dr. Tiago P. Peixoto 

Academic Editor

PLOS ONE